# Lactate in Sarcoma Microenvironment: Much More than just a Waste Product

**DOI:** 10.3390/cells9020510

**Published:** 2020-02-24

**Authors:** Maria Letizia Taddei, Laura Pietrovito, Angela Leo, Paola Chiarugi

**Affiliations:** 1Dipartimento di Medicina Sperimentale e Clinica, Università degli Studi di Firenze, Viale Morgagni 50, 50142 Firenze, Italy; 2Dipartimento di Scienze Biomediche Sperimentali e Cliniche, Università degli Studi di Firenze, Viale Morgagni 50, 50142 Firenze, Italy; laura.pietrovito@unifi.it (L.P.); angela91leo@gmail.com (A.L.); 3Tuscany Tumor Institute and “Center for Research, Transfer and High Education DenoTHE”, 50134 Florence, Italy

**Keywords:** sarcoma, lactate, microenvironment, acidity, immune response

## Abstract

Sarcomas are rare and heterogeneous malignant tumors relatively resistant to radio- and chemotherapy. Sarcoma progression is deeply dependent on environmental conditions that sustain both cancer growth and invasive abilities. Sarcoma microenvironment is composed of different stromal cell types and extracellular proteins. In this context, cancer cells may cooperate or compete with stromal cells for metabolic nutrients to sustain their survival and to adapt to environmental changes. The strict interplay between stromal and sarcoma cells deeply affects the extracellular metabolic milieu, thus altering the behavior of both cancer cells and other non-tumor cells, including immune cells. Cancer cells are typically dependent on glucose fermentation for growth and lactate is one of the most heavily increased metabolites in the tumor bulk. Currently, lactate is no longer considered a waste product of the Warburg metabolism, but novel signaling molecules able to regulate the behavior of tumor cells, tumor-stroma interactions and the immune response. In this review, we illustrate the role of lactate in the strong acidity microenvironment of sarcoma. Really, in the biological context of sarcoma, where novel targeted therapies are needed to improve patient outcomes in combination with current therapies or as an alternative treatment, lactate targeting could be a promising approach to future clinical trials.

## 1. Introduction

Sarcomas are rare malignancies (12% of all human cancers) of mesenchymal origin. Although there exist more than 70 subtypes, it is possible to divide them into two main categories: soft tissue sarcomas (STSs), which are derived from fat, muscle, nerve, sheath and blood vessels, representing less than 1% of all new cancer diagnoses, and bone sarcomas [1,2]. Among STSs, liposarcoma (LPS) and undifferentiated pleomorphic sarcoma are the most common subtypes in adults, while rhabdomyosarcoma is one of the highest incidences of tumors in children [3,4]. Concerning bone sarcomas, Ewing’s sarcoma (EWS) is one of the most frequently diagnosed in children together with osteosarcoma (OS), which is also a common subtype in adults as well as chondrosarcoma [5]. In addition to different histological genesis, sarcomas are characterized by high genetic instability and molecular heterogeneity [6]. To date, conventional therapies are based on surgical resection followed by radio- and chemotherapies, but mortality still remains unchanged, since sarcomas often develop recurrences and resistance to treatments. New diagnostic and therapeutic strategies are critically needed. 

Recently, studies on targeted therapies involving the tumor microenvironment (TME) are acquiring increasingly prominent. Indeed, the sarcoma microenvironment is a very complex and dynamic milieu, characterized mainly by high interstitial acidosis and high-density immune infiltrate. In this context lactate, the end-product of fermentative glycolysis released by both cancer cells and TME components is emerging as a key player of tumor progression, besides, its metabolic role is known, affecting the invasive abilities of cancer cells, the angiogenic process as well as the immune response [7].

High lactate levels and elevated immune infiltrate of the sarcoma microenvironment could be the basis for a remarkable source of therapeutic targets and future promising clinical trials.

## 2. Sarcoma Microenvironment

Cancer cells can educate the stromal cells resident in the microenvironment to its own benefits. The TME is characterized by a heterogeneous cell population that contributes to enhancing tumor growth, progression and aggressiveness by secretion of growth factors, hormones, cytokines and extracellular matrix and by an interaction between cell surface receptors and adhesive ligands [8,9].

The sarcoma microenvironment is a highly vascularized mesenchymal tissue, mainly represented by mesenchymal stem cells (MSCs), which exert a key role in sarcoma onset and progression [10,11], and by tumor-infiltrating immune cells, which affect cancer outcome and prognosis (Figure 1).

MSCs can differentiate towards diverse types of cell such as myofibroblast-like cells, pericyte-like cells, chondrocytes, adipocytes, osteocytes, cancer associated fibroblasts (CAFs) and tumor associated macrophages (TAMs). Increasing evidence suggest that MSCs might be the tumor initiating cells, namely the origin of a spectrum of sarcomas, both pleomorphic and translocation-driven subtypes, although there is still a great controversy about this aspect. Together with oncogenic events that occur during MSCs differentiation, it is well recognized a prominent role of the microenvironment that favors malignancies development, strengthening the “*seed and soil*” theory [12]. Previous studies have produced controversial results about the role of MSCs in tumor progression: especially whether MSCs support or suppress tumor growth. Despite this old dispute, it is now clear that systemically administered MSCs are recruited to tumors. Indeed, tumors and their TME prompt MSCs recruitment through different mechanisms that depend mainly on various inflammatory cytokines, chemokines and growth factors [13]. Moreover, several studies report that subsequently to MSC recruitment, tumor tissues educate MSCs to adopt a tumor-growth promoting phenotype [13]. In response to tumor derived acidity, resident MSCs are reprogrammed to tumor tissue-derived MSCs (t-MSCs), which further increase local acidification sustaining tumor progression [14,15]. Several studies demonstrated that MSCs induce pro-proliferative effects on tumor cells, promote OS stemness and epithelial to mesenchymal transition (EMT), recruit immunosuppressive cells and support tumor angiogenesis [16,17]. Our group has demonstrated that OS cells promote bone marrow derived-MSCs homing by production of monocyte chemoattractant protein-1 (MCP-1), growth-regulated oncogene-α (GRO-α) and transforming growth factor-β1 (TGF-β1). Once recruited inside the tumor lesion, in response to tumor-secreted cytokines, MSCs stimulate stemness, invasiveness and mesenchymal to amoeboid transition in OS cells, increasing their transendothelial migration. Furthermore, MSCs-stimulated OS cells increase the expression of pro-angiogenetic factors and promote migration, invasion and formation of the capillary network of endothelial cells in vitro [18]. Moreover, between MSCs and sarcoma cells a strict cross-talk is established: really, local tumor-derived acidosis, as well as, tumor associated osteolysis exert a great impact on MSC stemness [14,15]. Also lactate, the main driver of tumor acidosis, has a key role in OS progression: Bonucelli G. et al. demonstrated that MSCs are induced by adjacent OS cells to undergo Warburg metabolism and hence to increase lactate production and monocarboxylate transporter 4 (MCT4) expression. Thus, MSC-derived lactate feeds OS. Indeed, OS cells, through MCT1, import lactate, which drives mitochondrial biogenesis and promotes the migratory skill of OS cells. Probably, the role of lactate in this context is not only to fuel OS but rather to acidify the medium, thereby supporting a metastatic phenotype [19,20].

Confirming the role of MSCs in sustaining tumor progression, it has been demonstrated that in the sarcoma microenvironment, MSCs are more abundant than in adjacent normal tissue, this phenomenon is particularly evident in OS [21]. In keeping, in EWS tissues, tumor infiltrating MSCs are more proliferative than normal MSCs and have high expression of proliferation genes respect to normal MSCs [22].

Of note, MSCs exhibit unique immunomodulatory properties [23]. Through the secretion of paracrine factors, such as growth factors, cytokines and exosomes, or by cell-to-cell contacts, MSCs may affect the behavior of different immune cell types, including T and B-lymphocytes, monocytes, macrophages and dendritic cells [23,24,25]. MSCs are able to inhibit CD4^+^ and CD8^+^ proliferation, to decrease cytokines production by CD4^+^ and to suppress cytotoxic activity of CD8+ and natural killer (NK) cells [26]. Furthermore, MSCs stimulate polarization of T cells toward the regulatory phenotype, supporting their immunosuppressive functions [27,28]. It has also been proved that MSC-derived exosomes inhibit the activation and proliferation of B cells and reduce immunoglobulin secretion [29,30]. In addition, MSCs promote M2 macrophage polarization and inhibit chemotaxis of monocytes within the inflammatory lesion [23,31,32]. Finally, MSCs cooperate in the recruitment, maturation and function of dendritic cells, the most important antigen presenting cells in the body [33,34,35].

Consequently, by modulating the composition of the immune infiltrate, MSCs play a prominent role in many types of sarcoma. Indeed, several studies highlighted that the immune component of TME is crucial in determining the overall survival, prognosis and response to the therapy of several types of sarcoma. In EWS sarcoma, CD8^+^ cells infiltration correlates with improved survival [5]; also in several STSs including gastrointestinal stromal tumor (GIST), leiomyosarcoma, cutaneous angiosarcoma, high-grade undifferentiated pleomorphic sarcoma and synovial sarcoma, the presence of tumor-infiltrating lymphocytes correlates with improved prognosis [36,37,38]. In a mouse model of spontaneous GIST, the immune system cooperates substantially with Imatinib therapy. Indeed, Imatinib activates CD8^+^ T cells and induces regulatory T (Treg) apoptosis inside the tumor. Coherently, the CD8^+^ T cell to Treg ratio in human GISTs specimen with acquired resistance to Imatinib is significantly lower than in sensitive tumors [39].

Conversely to cytotoxic cells infiltrates, which exert a tumor suppressive action, the presence of TAMs and Tregs in the TME of different sarcoma types correlate with a worse prognosis. High density of both M2-polarized TAMs and Tregs has been shown in GIST, EWS, uterine and non-uterine LPS and myxoid LPS [40,41,42,43,44]. In OS, IL-34 production contributes to tumor growth by increasing the neo-angiogenesis and the recruitment of M2 macrophages [45]. Interestingly, in a xenograft human OS model, Xiao Q. et al. demonstrated that the recruited macrophages were polarized toward M2 subtype and tumor growth was strongly inhibited by the specific deletion of this cell population [46].

In addition, the expression on tumor cells of the checkpoint protein programmed death-ligand 1 (PD-L1) and the infiltration of programmed cell death protein 1 (PD1)-positive lymphocytes have been studied as a potential prognostic factor in sarcomas. In OS patients, the progression of the disease has been associated with the presence of the suppressive receptor PD1 on peripheral CD4^+^ and CD8^+^ T cells [47]. Furthermore, Kim C. et al. analyzed the relevance of the intra-tumoral infiltration of PD1-positive lymphocytes and PD-L1 expression in a cohort of 105 patients bearing of STSs. The presence of both intra-tumoral infiltration of PD1-positive lymphocytes and PD-L1 expression were significantly associated with higher clinical and histological stage, presence of distant metastasis and poor tumor differentiation [48]. Indeed, immune checkpoint inhibitors, which have emerged as encouraging therapy in other tumor types, are now tested in a range of sarcoma subtypes either as single agents or in combination with chemotherapy or kinase inhibitors [49]. 

The sarcomas TME is also rich with immunosuppressive cytokines including vascular endothelial growth factor (VEGF) that, together with hypoxia-inducible factor-1 α (HIF-1α), inhibits the maturation of dendric cells and promotes M2 macrophages and Treg migration inside the tumor stroma [50,51]. Indeed, in sarcomas high expression of VEGF and hypoxia correlate with poor prognosis and resistance to chemotherapy [52].

Moreover, high concentrations of indole 2,3-dioxygenase 1 (IDO1), which promotes the expression of kynurenine, thus stabilizing Tregs and suppressing the activity of cytotoxic T cells [53], correlate with lower metastasis-free survival and overall survival in sarcoma patients. In GST, Imatinib potentiates the anti-tumor T cell response through the inhibition of IDO [39].

In summary, these data show that most sarcomas exhibit immune cell infiltrates, but the tumor and immune microenvironment tends to be immunosuppressive. A deep investigation to underscore which kind of immune cells populate the TME and their role in tumor progression will be fundamental for prognosis and survival, to predict the response to immunotherapy and to develop new strategies to defeat tumors.

A second essential aspect of sarcoma microenvironment is the strong extracellular acidification. Indeed, deregulation of acidity, especially lactic acidosis, is crucial to promote tumor growth and metastasis, as already reported for sarcomas and other cancer types [15,54,55]. In this light, we first illustrate the main role of lactate as general driver of tumor progression based on the recent literature and then we deeply analyze its specific role in the field of sarcoma biology.

## 3. Lactate in Tumor Progression

In the 1920s, Otto Warburg described that ascite tumor cells consume great amounts of glucose and secrete high levels of lactate rather than completely oxidize it [56]. Indeed, high proliferative cells increase their glycolytic flux to generate ATP, most of the produced pyruvate is then transformed to lactate by lactate dehydrogenase A (LDH-A). The restoration of the NAD^+^ from NADH due to LDH-A activity allows the sustenance of glycolysis. In agreement, lactate derived by both stromal and/or tumor cells is one of the most upregulated metabolite across the TME [57,58] and LDH-A expression is induced by oncogenes such as Myc [59] and ErbB2 [60]. Mainly, this *“forced”* glycolysis flux and the disposal of great amount of lactate provides: i) intermediates for biosynthetic pathways and; ii) the acidification of the extracellular milieu (through lactate excretion) which impedes the development of a proper immune response, promotes invasion and metastasis of tumor cells. 

Lactate is transported across plasma membrane by a family of MCTs with different isoforms (MCT1–4). All of them require basigin (also known as CD147 or EMMPRIM) for their proper placement in the membrane. Even though all MCTs are bidirectional symports, MCT4 mainly facilitates lactate export while, MCT1 plays a key role in cellular lactate uptake. Many types of cancers overexpress both MCT1 and MCT4 as well as basigin [7]. It has been shown that MCT1 inhibition successfully prevents tumor cell growth [61,62,63,64]. Also MCT4 block, leading to acidosis of cancer cells, could be useful to halt tumor progression [63,65]. In agreement, CD147 silencing reduces pancreatic tumor malignancy both in vivo and in vitro [66,67] and CD147 gene ablation leads to a downregulation in MCT1 and MCT4 expression and to a consequent decrease of lactate export in non-small cell lung cancer (NSCLC) [68].

Recently, it has been demonstrated that carbonic anhydrases (CAs), key regulators of intracellular and extracellular acidity, facilitate lactate and H^+^ transport across MCTs by a mechanism independent from their enzyme catalytic function [69]. Really, CAs function as “proton antenna” for the transporters: intracellular CAII collects H^+^ from the surroundings and donates them to the transporters. On the extracellular side instead, CAIX can remove H^+^ from the transporter and then transfers it to the adjacent protonable residues. This mechanism is particularly efficient in hypoxic cancer cells producing high levels of lactate and H^+^, which have to be removed from the cytoplasm to avoid intracellular acidosis [69,70]. In keeping, antibodies directed against CAIX results in a substantial decrease of lactate export and in a consequent reduction of cancer cell proliferation [70].

Lactate is also a respiratory substrate and a lipogenic precursor for some cancer cell types [71]. Recent evidences show that, in human NSCLC, the in vivo contribution of lactate to the tricarboxylic acid cycle (TCA) predominates the glucose one [58]. Moreover, Sonveaux P. et al. demonstrated that there is a close symbiosis between glycolytic and oxidative tumor cells: indeed, lactate derived from hypoxic tumor cells diffuses to oxygenated tumor ones, which imports and oxidizes the molecule to produce energy. Really, this metabolic symbiosis could be destroyed through the inhibition of the transporter MCT1 [61]. Interestingly, Lisanti’s group coined the expression “The Reverse Warburg Effect” to describe the uptake of energy rich metabolites by cancer cells to sustain TCA cycle and ATP production. They showed that epithelial cancer cells promote the aerobic glycolysis in neighboring stromal fibroblasts. In turn, these CAFs produce lactate and pyruvate [72]. Finally, cancer cells could upload these energy-rich metabolites and use them in the mitochondrial TCA cycle, thereby supporting efficient energy production [72,73] and sustaining tumor growth and metastasis. Indeed, Bonucelli et al. showed that exogenously added lactate can promote cell migration and fuel lung metastasis in a model of MDA-MB-231 breast cancer xenografts. Moreover, the same authors discovered that in human breast cancer samples, TCA cycle and mitochondrial metabolism are upregulated in tumor epithelial cells, in comparison to the adjacent stromal cells [72]. 

In this scenario, our group has demonstrated that CAF-derived lactate is uploaded by neighboring prostate cancer cells for anabolic purposes. Inside tumor cells, lactate oxidation to pyruvate alters the NAD+/NADH ratio, thereby activating the NAD^+^-dependent deacetylase Sirtuin1/PGC-1α axis and potentiating mitochondrial metabolism of prostate cancer cells and their invasive abilities [74]. 

Collectively, it is possible to assume that lactate is a highly produced metabolite in the tumor milieu, being secreted by several different types of cancer, as well by hypoxia exposed cancers. In addition, also stromal cells like CAFs, TAMs and effector T cells are sources of lactate. Thus, as mentioned above, although the centrality of the Warburg metabolism has been confirmed to sustain tumor behavior, it is likely that increased levels of lactate in tumor milieu are essential also to directly modulate stromal cells and their interplays with tumor cells, deeply affecting the tumor progression (Figure 1). 

We now review the main roles exerted by lactate in promoting a pro-tumoral phenotype of cancer and stromal cells.

### 3.1. Lactate and Migration

Increased lactate levels correlate with amplified metastatic potential in various human primary carcinomas [75]. In glioma cells, lactate induces TGFβ-2 expression, thus stimulating the expression, secretion, and activation of matrix metalloproteinase-2. These findings highlight a crucial mechanism for glioma invasion, based on lactate production [76]. Moreover, it is widely accepted that exogenous addition of lactate increases cell motility of different tumor cell lines [77]. In keeping, Ippolito L. et al. demonstrated that there is a strict correlation between CAF-induced lactate addiction and EMT activation in prostate cancer cells [74]. Furthermore, lactate is now considered an angiogenic promoter and a clear relationship between lactate and endothelial cell migration has been established, too. 

Lactate stabilizes HIF-1α protein: indeed, lactate can be converted, by LDH-B, into pyruvate, an inhibitor of HIF-prolyl-hydroxylases (PHDs). In turn, PHD2 impairment leads to increased stability of HIF-1α, which stimulates VEGF accumulation in tumor cells [78,79]. Accordingly, lactate-induced HIF-1α activation in normoxic endothelial cells causes an increase in vascular endothelial growth factor receptor 2 (VEGFR-2) and basic fibroblast growth factor expression, thus promoting endothelial cell migration and vascular sprouting. In keeping, MCT1 inhibition blocks tumor angiogenesis [80]. Moreover, lactate is able to induce in endothelial cells another pro-angiogenic factor, namely interleukin-8, which sustains new blood vessels maturation and increased cell migration [81]. In addition. Lee D.C. et al. reported a direct role of lactate in modulating angiogenesis independently of HIF-1α in different cell lines. Indeed, in hypoxic condition N-Myc downstream-regulated gene 3 (NDRG3) is stabilized by its direct binding to lactate, which protects NDRG3 by PHD2/VHL axis-mediated degradation. Stabilized NDRG3 protein activates the Raf/ERK pathway and promotes angiogenesis, cell growth and tumor formation [82]. 

### 3.2. Lactate and Resistance to Therapy

Of note, lactate contributes heavily to acidification of the TME and to chemoresistance. Several drugs, including chemotherapeutics, are weak bases and the acid milieu easily impairs both the uptake by tumor cells and reduces their efficacy [83,84].

Recently Apicella M. et al. demonstrated that lactate, through the induction of CAF-derived hepatocyte growth factor (HGF), sustained resistance to MET/EGFR tyrosine kinase inhibitors, activating MET-dependent signaling in cancer cells. In agreement, the targeting of molecules involved in the lactate axis (such as LDH, MCT1 and MCT4) abolished in vivo resistance, suggesting a key role of lactate in this phenomenon [85].

Additionally, lactate levels correlate also with radioresistance of many cancers [86]. Sattler U.G. et al. demonstrated that in human head and neck squamous cell carcinoma (HNSCC) xenografts, lactate concentration positively correlates with tumor resistance to fractioned irradiation [87]. Moreover, Koukourakis M.I. et al. proved that also LDH5 overexpression can be associated with radioresistance of HNSCC, prostate and bladder cancers [88,89,90]. At least in part, this effect could be due to the antioxidant properties of lactate as shown by its ability to prevent lipid peroxidation. Most likely, lactate could be used as a predictive molecule to hypothesize the therapeutic responses of tumor [91].

### 3.3. Lactate and Transcription

Recently, the metabolic state of the cell has been linked to gene transcription modification, too. Interestingly, Latham T. et al. showed that lactate is an endogenous histone deacetylase (HDAC) inhibitor, even if less efficient than established HDAC inhibitors, such as trichostatin A and butyrate. All these compounds induce deregulated expression of the same genes (even if with a different extent), suggesting a common mechanism of action [92]. Really, the concept of lactate uploaded for only cellular needs seems to become reductive. Both in MCF7 and L6 cells, exposure to lactate profoundly alters the transcriptional profile, upregulating respectively 4131 and 673 genes, including MCT1 [93,94]. Moreover, also HIF-1α levels are stabilized by lactate exposure in different tumor cell types [78,79]. Furthermore, lactate-induced HIF-2α -c-Myc signaling stimulates the glutamine pathway by triggering the expression of both glutamine transporter ASCT2 and glutaminase-1. Since MCT1 silencing hampers lactate-induced c-Myc activation, MCT1 inhibitors could be useful to indirectly impede also glutamine uptake and metabolism in oxidative cancer cells [95]. 

A striking novel function of lactate in transcriptional control has been recently elucidated [96]. Zhang D. et al. described a lactate-derived lactylation of histone lysine residues as an epigenetic modification able to promote gene transcription. In particular, increased histone lactylation during M1 macrophage activation stimulates homeostatic genes promoting a late-phase switch to an M2- like phenotype [97].

### 3.4. Lactate and Signal Transduction

To corroborate the fact that lactate is not only a metabolic substrate, but also a signaling molecule able to regulate gene expression and protein activation, it is now known that lactate can bind to a specific receptor, named G protein-coupled receptor 81 (GPR81), a lactate-activated G-protein-coupled receptor. Physiologically, GPR81 activation reduces lipolysis through the decrease of cAMP levels in adipose cells [98]. Also different tumor cell lines such as colon, hepatocellular, lung, pancreatic ductal adenocarcinoma (PDAC) and so on, express GPR81 which contributes to modulate tumor cell response to lactate levels [99,100]. The lactate/GPR81 axis in PDAC increases the expression of MCT1-4, CD147 and PGC-1α, promoting lactate uptake and oxidative metabolism [99].

Recently, a key role of GPR81 increase in the development of chemoresistance has been elucidated in cervical carcinoma HeLa cells via ABCB1 transporter upregulation [101]. Moreover, several evidences show that lactate-mediated GPR81 activation sustains lactate upload by tumor cells and angiogenesis. Indeed, breast tumor xenografts silenced for GPR81, show an impaired cancer growth and a suppressed angiogenesis, due to a decreased production of the pro-angiogenic mediator amphiregulin [102]. Interestingly, lately it has been revealed a new role of lactate in osteoblast differentiation. In particular, Yu Wu et al. demonstrated that in response to lactate exposure, the activation of the GPR81-PKC-Akt signaling, positively affects the PTH-mediated osteoblast differentiation, independently from MCT1 function [103].

### 3.5. Lactate and Immunomodulatory Role in TME

Metabolic features of TME, low glucose and high lactate concentrations, actively regulates the reactivity and functionality of different immune infiltrating components (Box 1) [104].

Box 1Key immuno-players in sarcoma TME.
**CD8^+^ T cells** directly exert cytotoxic effects on tumour cells by recognizing tumor-associated antigens. Increase of CD8^+^ in TME infiltrating is correlated with better prognosis [105]**CD4^+^ T cells** may differentiate into:-T cell effector (Teff);-T cell regulatory (Treg).Teff and Treg require distinct metabolic programs to exert their function: of note, lactate accumulation and acidosis
are two determinant features of TME affecting T cells behaviour.CD4^+^ T cells may, in turn, activate or inhibit other downstream immune cells, modulating the immune response against cancer [106]**TAMs** are the main component of immune infiltrate in several tumours. Inside TME, macrophages may polarize to a pro-inflammatory (M1) phenotype with tumoricidal activity, or toward an alternatively activated phenotype (M2), that plays an intriguing role in tumour [107]. Really, M1 macrophages are significantly dependent on glycolysis and secrete high amount of lactate. Conversely, M2 macrophages show enhanced mitochondrial oxidative phosphorylation to sustain their metabolism [108]**NKs T cells** are innate immune system cells characterized by granules which contains granzymes (proteolytic enzymes) and perforin (protein which break the cell membrane) used to defence the body against tumour and virus and lead the death of target cells by apoptosis [109]**MSCs** are the most represented stromal cells within sarcoma TME. MSCs exhibit remarkable immunomodulatory properties, being able to affect the recruitment, activation and function of T and B cells [23].



Indeed, the lactate is emerging as central modulator of immune and inflammatory responses occurring during tumor growth (Figure 2).

#### 3.5.1. Lactate and T cells

In lactate-rich environment, Teff, as well as CD8^+^ cells, become anergic since they depend on glucose to proliferate and perform their effector functions [110,111]. It has been proposed that competition between tumor cells and Teff for glucose within TME could contribute to establish an immunosuppressive environment [110]. Conversely, Angelin A. et al. demonstrated that the Treg transcription factor Foxp3 reprograms T cell metabolism, potentiating oxidative phosphorylation and suppressing Myc and glycolysis [112]. In line, Comito G. et al. showed that CAFs-produced lactate reduces CD4^+^ Th1 population by SIRT1-mediated degradation of T-bet, and stimulates Treg proliferation by promoting FoxP3 activation in a prostate cancer model [113]. In addition, lactate acidosis has been negatively linked to T cell trafficking. For instance, it has been demonstrated that through specific membrane carriers selectively expressed on CD4^+^ and CD8^+^ T cells, lactate provides a stop signal, entrapping T cells in inflammatory sites [114]. In murine models of melanoma, tumors with reduced lactate accumulation (LDH-A ^low^) show increased T cells infiltration and slower growth compared to control group [115]. Again, lactic acid may suppress the proliferation and cytokine production of CD8^+^ T cells, reducing their cytolytic activity [116] and overexpression of Glut-1 within tumor tissue inversely correlates with CD8^+^ infiltration [117] and survival in certain cancers [118]. Notably, in a mouse models of melanoma, combining immunotherapy (anti-CTLA-4 and anti-PD1) with oral bicarbonate buffer therapy reduces tumor growth and increases T-cell trafficking inside cancer lesion [119]. In line, Cascone T. et al. recently demonstrated that tumor glucose metabolism is a mechanism of resistance to T-mediated anti-tumor immunity and suggest targeting glycolysis pathway in combinatorial therapeutic intervention in melanoma patients [120]. 

#### 3.5.2. Lactate and Macrophages

In mouse models of breast cancer, hypoxia and lactate orchestrate macrophages polarization and endothelial cells activation, promoting the neovascularization process in ischemic regions [121]. In syngeneic murine models of Lewis lung carcinoma and melanoma, lactate released by tumor cells drives macrophages polarization toward M2 phenotype and stimulates VEGF production via HIF-1α [122]. In line, it has been proved that in HNSCC model, tumor-secreted lactate inhibits macrophages infiltration in TME and stimulates M2 polarization [123]. Moreover, in monocytes and macrophages, lactate suppresses TLR4-mediated release of IL-1β via GPR81 [124]. 

#### 3.5.3. Lactate and NKs

It has been reported that lactate inhibits expression of perforin and granzymes in both human and murine NKs in vitro [125] and LDHA^low^ tumors show reduced infiltrating NKs [115]. Furthermore, LDH-associated lactic acid increase inhibits the upregulation of the transcription factor NFAT in T and NKs cells resulting in decreased interferon-γ production [115]. Along this line Dichloroacetate, an inhibitor of pyruvate dehydrogenase kinase and hence a suppressor of glycolysis, by reducing acidity, increases NKs and CD8^+^cells infiltration in tumor-bearing mouse spleen [125].

Overall, all these data support a wide-broad spectrum role for lactate shuttle, supported by its transporters and by GPR81, that covers from the metabolic adaptations to cell migration, transcriptional processes and the immune response, far from its long-recognized role as a simple waste product of the glycolytic pathway (Figure 3).

## 4. Lactate and Sarcomas

As already mentioned, abnormal glucose metabolism characterizes many tumor cells. In agreement, since 1988 it has been described that patients with different sarcoma types showed an increase in glucose uptake and a decrease in its oxidation, attesting an altered glucose metabolism [126]. The enhance of the glycolytic flux guarantees key benefits to cancer cells with respect to mitochondrial oxidation: first, a faster ATP production, which confers a growth advantage to cancer cells; second, the production of glycolytic intermediates to fuel divergent pathways which satisfy the metabolic demands of proliferating cells [127]. Proliferative cells gain several molecular adaptations to sustain this high glycolytic flux, especially modifying the expression/activity of different glycolytic enzymes. It has been reported an increased expression of phosphofructokinase-2 (PFK2), which produces fructose-2,6-bisphospate, an allosteric activator of PFK1, the main driver of the glycolytic flux [128,129]. Moreover, proliferative cells express higher levels of pyruvate kinase M2 (PKM2) which can be inhibited by allosteric and covalent modifications [130,131]. PKM2 inhibition creates a metabolic bottleneck at the end of glycolysis, sustaining the accumulation of upstream glycolytic intermediates, which are then channeled into synthetic processes to produce nucleic acids, phospholipids and amino acids, greatly needed by highly proliferating cells, such as tumor ones [132]. Finally, several cancer cells upregulate LDH which generates NAD^+^ from NADH to sustain the glycolytic flux [59]. 

Recently, a specific involvement of the gluconeogenic enzyme fructose-1,6-bisphosphatase2 (FBP2) has been shown in STS’s progression. Indeed, FBP2 is strongly downregulated in different types of sarcoma, while FBP2 re-expression in sarcoma cells severely inhibits tumor cell growth both in vitro and in vivo, showing a possible tumor-suppressive role of FBP2 [133]. The authors point out that FBP2 decrease leads to the increased glycolytic activity observed in sarcomas, while FBP2 re-expression hinders tumor progression through two mechanisms: i) inhibition of glucose uptake and lactate secretion; ii) restrain mitochondrial biogenesis by repressing c-Myc-dependent transcriptional activity [133].

In agreement with increased glucose utilization by cancer cells, the [18F]-fluorodeoxyglucose (FDG) positron emission tomography (PET), based on the uptake of labeled glucose analog by tumor cells, is a powerful non-invasive imaging modality that correctly predicts histopathologic response in several malignancies. It has been reported that a correlation between FDG-PET changes and histologic response is present in pediatric OS and EWS, too [134]. Moreover, in a recent paper Schmidkonz C. et al. demonstrated that an integrated analysis of FDG-PET/computed tomography (CT) and circulating tumor DNA (ctDNA) quantification is a potent instrument to evaluate responses to multimodal chemotherapy and to detect tumor relapses in EWS patients [135].

Another recent manuscript correlates highly in vivo tumorigenic OS cells with great dependence on glycolysis and with the suppression of OXPHOS compared to low tumorigenic ones. In agreement, glycolytic inhibitors combined with chemotherapy reduce tumorigenesis of OS cells, suggesting that glycolysis inhibition combined with chemotherapy might be a promising adjuvant treatment for OS [136].

Specifically, regarding lactate, recent evidence suggests a role of this metabolite also in the progression of different types of sarcoma. 

As mentioned, human LDH-A has been found overexpressed in various cancer tissues, including sarcomas [137,138]. Genetic silencing of LDH-A caused a reduced OS cell proliferation and migration in vitro, and in impaired tumorigenesis in vivo [139].

Fu Y. et al. performed a meta-analysis to evaluate the prognostic role of serum LDH in 2543 OS patients. The authors established that elevated serum LDH levels correlate with both lower event-free survival and overall survival, suggesting LDH as a prognostic biomarker for OS patients [140]. A similar meta-analysis study conducted on 943 OS patients showed that the LDH level is associated with a lower overall survival rate in OS patients [141]. A strict correlation between high levels of serum LDH and tumor progression has been demonstrated also in patients with EWS [137]. This last meta-analysis demonstrates that high levels of serum LDH are associated with lower overall survival and 5-year disease-free survival rates in patients with EWS, suggesting that serum LDH levels should be considered as a new biomarker for EWS prognosis. Also, LDH-B, which converts lactate in pyruvate to be then diverted into TCA for energy purpose, is highly expressed in OS cell lines and predicts a poor prognosis in patients. LDH-B silencing inhibits cell growth and proliferation [142]. Conversely, although elevated LDH levels have been recognized as a negative prognostic factor for several solid tumors, hematological malignancies and sarcomas, a recent univariate and multivariate analysis conducted on 142 adult patients with primary STS showed that elevated LDH levels previous to treatment were not a poor prognostic factor for oncological outcome including disease-specific survival and event-free survival, at least for this subset of sarcomas [143].

Several LDH inhibitors (LDHi) are known, such as oxamate or gossypol, but due to their weak activity or high toxicity, novel compounds are currently identified to inhibit LDH. The efficacy of some of them has been proved in sarcoma cells. In a recent paper, by computer-aided virtual screening approach, Cao W. et al. discovered a potent inhibitor of hLDH5 able to decrease the growth of MG-63 cells [144]. Also, Fang A. et al. discovered a new compound, through a docking-based virtual screening, able to target LDH-A. This compound, by blocking LDH-A, induces apoptosis of MG-63 cells in a dose-dependent manner, downregulates lactate production and acidification, and promotes upregulation of oxygen consumption rate [145]. Interestingly, Yeung C. et al. demonstrated that the oncogenic transcription factor EWS-FLI1, which acts as the primary driver of EWS, increases the expression of LDH-A; accordingly, EWS-FLI1 silencing induces a decrease in LDH-A protein, in four different EWS cell lines. These authors clearly demonstrated that EWS cells depend on LDH-A expression and genetic or pharmacologic inhibition of LDH-A reduces glycolysis, cell growth and promotes apoptosis in EWS cells both in vitro and in vivo. Indeed, in vivo treatment with LDHi, such as the NCI-737 compound, is able to considerably reduces the xenograft growth [146]. However, only in mice treated for long-term with frequent intratumor injections of high doses of the drug (75 mg/kg) a significant reduction of tumor growth is observed. Unfortunately, the anti-tumor effect of treatment with LDHi was accompanied by systemic toxicity (hemolytic anemia) [146]. Nevertheless, targeting LDH-A activity might represent a promising therapeutic strategy for sarcoma treatment. 

In addition, a role for LDH-A in the resistance of EWS to Cetuximab has been established. LDH-A expression is upregulated in Cetuximab-resistant EWS tissues and cell lines. Moreover, the authors demonstrated that inhibition of LDH-A with different approaches re-sensitized resistant cells to Cetuximab [147]. Also in chondrosarcoma cell lines, elevated LDH-A expression and high glycolytic flux have been associated with Doxorubicin resistance [148]. Indeed, combined treatment with Doxorubicin and oxamate displayed a synergistic effect on tumor cell viability both in vitro and in vivo. Notably, the downregulation of LDH-A by siRNA, re-sensitized tumor cells to the chemotherapeutic agent [148].

In keeping with the role of LDH in sarcoma’s progression, several microRNAs (miRNAs) targeting LDH have been found altered in different sarcomas. miR-33b has been found significantly downregulated in OS tissues with respect to adjacent non-tumor tissues. Moreover, overexpression of miR-33b or inhibition of LDH-A significantly suppressed glycolysis and proliferation of OS cells, demonstrating that LDH-A is a direct target of miR-33b and suggesting, once again, a central role of lactate in sarcoma progression [149]. Similarly, also miR-323a-3p targets LDH-A and exerts a suppressive role of glycolysis and OS growth. In agreement, miR-323a-3p results significantly downregulated in OS tissues and cell lines, while its overexpression reduces cell viability [150]. In addition, a different miRNA, namely miR-186, which decreases indirectly lactate levels through direct inhibition of HIF-1α and glycolysis, functions as a tumor suppressor of both HOS and U2 OS cell lines, inhibiting tumor cell proliferation and invasion [151].

Also in EWS, miR-34a, which targets LDH-A was found downregulated with respect to normal tissue or MSCs [152]. Furthermore, miR-34a expression has been found significantly lower in metastases compared to primary tumors. Once again, these authors demonstrated that low expression of LDH-A, due to high expression of miR-34a in EWS, was associated with a better event-free and overall survival [152]. 

It is now widely accepted that long non-coding RNAs (LncRNAs), controlling metabolic pathways, are involved in several sarcoma developments, including OS, EWS and GISTs, influencing cell proliferation, metastasis and drug resistance. LncRNA PVT1 is increased in OS cells and tissues and its expression correlates with poor prognosis. LncRNA PVT1 stimulates glucose uptake, lactate production and tumor progression acting as a molecular sponge to repress miR-497, which directly targets hexokinase 2 in OS [153]. Conversely, the expression levels of LncRNA HAND2-AS1 were significantly lower in OS tissues compared with healthy adjacent tissues [154]. LncRNA HAND2-AS1 controls energy metabolism and OS progression, indeed, HAND2-AS1 downregulation promotes cell growth, glucose uptake and lactate production through HIF-1α –activation [155]. 

We have already mentioned the correlation between lactate and reverse Warburg in OS cells, as shown by Bonuccelli G et al. Gorska-Ponikowska M. et al. showed that lactate administration induces both proliferation and migration of OS HOS-143B cells, while a potent anticancer agent, namely 2-methoxyestradiol (2-ME), currently under investigation in clinical trials for treatment of OS, breast and prostate carcinomas, is able to reverse these effects [156]. In particular, through the generation of nitro-oxidative stress and by inhibiting the mitochondrial respiratory complex I, 2-ME repressed lactate-induced cancer cell migration and promoted apoptosis in OS cells [156].

As mentioned, lactate secretion contributes to the increase in tumor extracellular acidification, which sustains both tumor progression and inhibition of a proper immune response. Elevated interstitial acidosis is a common feature of bone sarcomas microenvironment and it plays a key role in driving MSCs activation and stemness [14,15]. It has been shown that extracellular acidosis (with pH ranging from 6.5 to 6.8) maintains MSCs in the quiescent G0 phase impairing osteogenic differentiation, promotes the expression of stemness-related genes, such as OCT4 and SOX2 and increases MSCs clonogenic potential [14]. In this acid milieu, lactate-activated MSCs produce a plethora of signaling molecules, such as growth factors, chemokines and cytokines that affect tumor cell behavior [15]. In acidity-activated MSCs and OS cells co-cultures, tumor cells show enhanced proliferation, migration and invasion abilities compared to OS cells maintained with n-MSCs. Furthermore, the conditioned media from acidity-activated MSCs promote OS stemness and chemoresistance [15]. In keeping with this evidence, it has been shown that proton pump inhibition has a powerful anti-tumor effect against different tumor types [157,158]. The H^+^-rich milieu that characterizes the sarcomas microenvironment plays an intriguing role in driving tumorigenesis and in promoting chemoresistance. Indeed, tumor acidification is a mechanism by which tumor cells develop resistance to cytotoxic drugs. Concerning the typical acidic extracellular pH (pHe) of sarcomas, Ferrari S. et al. demonstrated that proton pump inhibitors (PPIs) sensitize both human OS cell lines and xenografts to chemotherapeutic treatment. Indeed, PPIs significantly increase the sensitivity of OS cells to cisplatin, both in vitro and in xenograft models, providing evidence for the efficacy of combined treatment of conventional chemotherapy and PPIs [159]. In addition, Avnet S. et al. showed that in OS cells, acidification promotes resistance to several chemotherapy agents, conversely, the addition of omeprazole, a PPI targeting lysosomal acidity, significantly increased the sensitivity to chemotherapeutic agents [160]. More recently, the same group published that in OS and EWS extracellular acidity is fundamental to support tumor cell survival. Indeed, a clear role for the proton pump V-ATPase has been found [161,162]. These authors demonstrated that acidity activates a stress-regulated switch through the pro-survival signaling cIAP/TRAF/NF-κB pathway [162]. Actually, V-ATPase is highly expressed in OS, rhabdomyosarcoma and chondrosarcoma, all sarcoma able to survive in the strong acid microenvironment. V-ATPase silencing and treatment with the PPI Esomeprazole interfere with the acidification process and induce a decrease in cell viability [163]. Unfortunately, this experimental evidence can be hardly translated to the clinic. For instance, recent clinical trials, which compare patients treated with Pazopanib an orally administered, potent tyrosine kinase inhibitor, approved for the treatment of advanced STSs with Pazopanib plus gastric acid-suppressive (GAS) agent, pointed out that co-administration of GAS agents with Pazopanib was associated with shortened progression-free survival and overall survival. The failure of the combined therapy may be due to the negative effect exerted by GAS agents on Pazopanib, which is practically insoluble at pH > 4 [164].

In conclusion, further preclinical researches are critically needed to deep scientific knowledge on pharmacokinetics, bioavailability and toxicity of LDHi, which might represent a new class of innovative anticancer treatment for sarcoma.

## 5. Conclusion

Alteration in acid/base homeostasis is one of the main drivers of tumor outcome. Both lactic and carbonic acidosis promotes microenvironmental acidification, which directly sustains tumor growth and metastasis. Of note, tumor acidity is detrimental for the immune components, contributing to immune escape and cancer progression, thus limiting the efficacy of immune checkpoint inhibition in different types of cancers [55,165,166].

In this light, a dual approach that simultaneously targets lactic acidity and promote the immune response could be truthful to fight tumor progression. Really, several pieces of evidence show that inhibition of lactate transport through MCTs block are effective to halt the growth of different types of tumor. In addition, more recently, it has been reported that also inhibition of CAIX, which facilitates the extrusion of protons and lactate through the MCTs, causes a decrease in cancer cell growth.

On the other side, modern immunotherapy wishes to counteract tumor evasion mechanisms through innovative immunotherapeutic strategies. Indeed, the development of monoclonal antibodies to inhibit immune checkpoints and the use of adoptive transfer of T cells transduced with chimeric antigen receptors (CARs) have been proven to be beneficial in a variety of cancers, including sarcomas [167,168,169]. Indeed SARC028, the first Phase II clinical study of 80 patients based on the administration of Pembrolizumab, an anti-PD-1 antibody to patients with advanced sarcomas has already provided results [170], while a HER2 s-generation CAR-T cell approach in patients with HER2^+^ OS and EWS unlighted the feasibility and safety of this treatment in patients with recurrent or refractory sarcomas [169]. Moreover, in clinical practice, a different immunotherapy strategy, based on the immune-stimulant Mifamurtide, which targets macrophages and monocyte, has been approved by EMA in 2009 for the treatment of high-grade non-metastatic OS in combination with postoperative chemotherapy [171]. Really, the inefficacy of Mifamurtide treatment in metastatic OS could be due to the differences in the immune infiltrates between primary and metastatic lesions.

Crucially, the application of immunotherapy as a promising effective strategy against sarcoma progression is growing. Current efforts are focused on identifying the different components of the immune microenvironment infiltrating sarcomas in order to draw a conclusion about prognosis and the choice of the proper immunotherapeutic treatment to apply.

In this scenario, pre-clinical studies showed the relevance of combined therapies based on inhibition of lactate release, or generic anti-acidic strategies, associated with immune therapies to counteract tumor progression [119,125,172,173]. 

In particular, due to the high lactic acidity microenvironment of sarcomas, this dual approach could be a very promising therapy to hinder these malignancies, which are very often unresponsive to standard radio- and chemotherapy.

## Figures and Tables

**Figure 1 cells-09-00510-f001:**
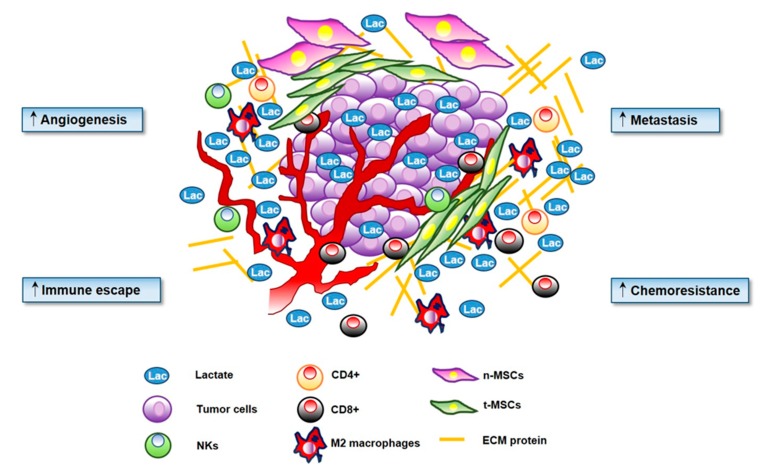
Lactate has a key role in cancer progression. Microenvironmental secreted lactate increases angiogenesis, motility and migration of cancer cells. Lactate is directly involved in the ‘immune escape’ by decreasing activation of T cells and promoting Treg proliferation. Lactate increases extracellular acidosis of the tumor microenvironment (TME) stimulating chemoresistance.

**Figure 2 cells-09-00510-f002:**
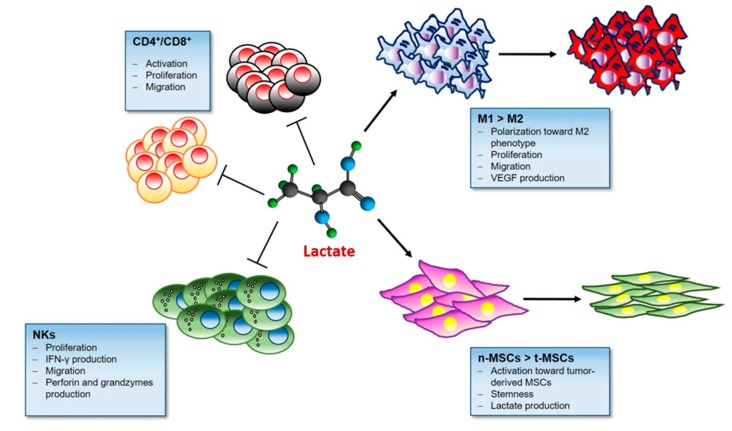
Lactate and the Immune response. Lactate in the tumor microenvironment impairs immune surveillance by blocking natural killer (NK) cells and tumor infiltrating T cells. Lactate-rich milieu promotes Treg cell survival and their immunosuppressive function. Moreover Lactate stimulates the M2 pro-tumoral polarization in macrophages and activates mesenchymal stem cells (MSCs).

**Figure 3 cells-09-00510-f003:**
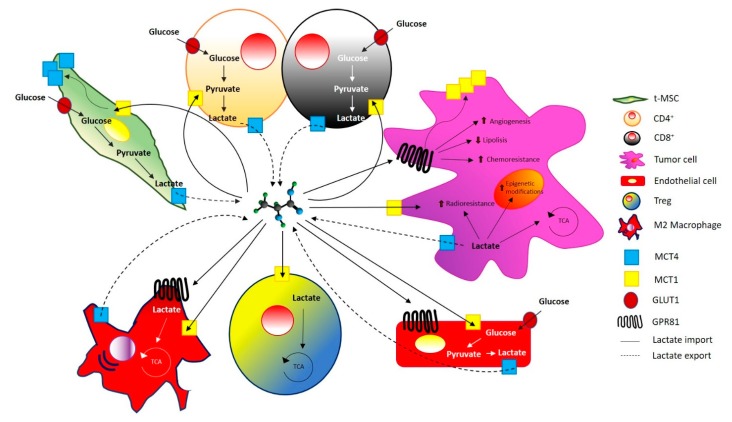
Lactate trafficking in tumor and stromal cells. Lactate shuttles through monocarboxylate transporter (MCT) present both in tumor and stromal cells. GPR81, a specific receptor for lactate is expressed on stromal cells and on different cancer cells, mediating chemoresistance, angiogenesis, decreasing lipolysis and up-regulating MCT1 expression.

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
