# Peer review of "Lactate in Sarcoma Microenvironment: Much More than just a Waste Product"

_cells, 2020, doi:10.3390/cells9020510_

Round 1
Reviewer 1 Report
In this interesting review paper Taddei and coworkers discuss recent studies supporting the view that lactate cannot be longer considered as a simple a waste product of the Warburg metabolism, but a novel signaling molecule able to regulate the behaviour of tumor cells, tumor-stroma interactions and the immune response. In this review, the role of lactate in the strong acidity microenvironment of sarcoma was illustrated. These studies have represented the basis for the development of new experimental therapies attempting to target some metabolic vulnerabilities of sarcoma cells.
This paper is mainly based on the analysis of the effects of increased lactate production in the tumor microenvironment and provides an exhaustive overview of this topic.
The impact of the paper could be improved by providing a brief overview of the glycolysis pathway.
The following paper need to be analyzed and discussed in this review:
1)Huangyang P, et al. Cell Metab. 2020; 31, 174-188.
2)Nakamura T et al. Anticancer Res 2019; 39, 6871-6875.
3)Mizushima et al. Cancer Sci. 2020; 111, 36-46.
4)Schmidkonz et al. Eur J Nucl Med Mol Imaging 2020, in press.
Author Response
We thanks the reviewer 1 for her/his positive comments to the manuscript.
As suggested, we added a brief overview of the glycolysis pathway (pag 18) and we inserted and discussed the suggested references: Huangyang et al.; Mizushima et al.; Schmidkonz et al., (pagg 18-19) and Nakamura et al. (pagg 19-20)
Reviewer 2 Report
The manuscript reads well and the topic described in the manuscript is important to further our understanding of tumor biology. Provision of an additional figure that shows functional relationship of MCT1, MCT4 and GPR81 with lactate in TME would strengthen the manuscript. Addition of concise description on lactate in Introduction would facilitate readers to preliminary figure out the topic of the manuscript.Author Response
We thanks the reviewer 2 for her/his positive comments to the manuscript.
As suggested by this reviewer, an additional figure (Fig.3), showing the relationship between MCTs and GPR81 with lactate in TME, has been included.
In addition, as suggested, a concise description of lactate has been added to the Introduction